# Precision medicine in laryngeal cancer: protocol of the laryngeal cancer cohort (LARCH)

David Winston Hamilton ©,[1,2] James O'Hara,[1,2] Amarkumar Rajgor,[1,2] Gerald Selby,[3] Mhairi Anderson,[4] Kim Keltie,[5,6] Rosalyn Parker ©,[5] Dawn Teare,[1] Joanne Patterson,[7] Terry M Jones,[7] Linda Sharp ©[1]

¹Population Health Sciences Institute, Newcastle University, Newcastle upon Tyne, UK
²ENT, Newcastle Upon Tyne Hospitals NHS Foundation Trust, Newcastle Upon Tyne, UK
³Patient representative, Newcastle upon Tyne, UK
⁴Newcastle Joint Research Office, Newcastle Upon Tyne Hospitals NHS Foundation Trust, Newcastle Upon Tyne, UK
⁵Northern Medical Physics and Clinical Engineering, Newcastle Upon Tyne Hospitals NHS Foundation Trust, Newcastle Upon Tyne, UK
⁶Translational and Clinical Research Institute, Newcastle University, Newcastle upon Tyne, UK
⁷Liverpool Head and Neck Centre, University of Liverpool, Liverpool, UK

**Correspondence to**
David Winston Hamilton;
david.hamilton@ncl.ac.uk

## ABSTRACT

**Introduction** Laryngeal cancer disproportionately affects socioeconomically disadvantaged patients. Treatment can render a patient nil by mouth or in need of a permanent tracheostomy. In the past 30 years, survival has remained at best static and at worst it has declined. Currently, there is no method of prognosticating how a patient will respond to treatment.

The LARyngeal Cancer coHort (LARCH) aims to establish how survival and quality-of-life outcomes compare between surgery and (chemo)radiotherapy in early and advanced laryngeal cancer and how the presenting features of laryngeal cancer influence oncological, functional and quality-of-life outcome.

**Methods and analysis** This study is the first enhanced laryngeal cancer disease cohort. In the initial phase, we aim to deliver a prospective cohort study of 150 patients in 8 centres over a 3-year period.

Patient, tumour, quality-of-life and laryngeal functional data will be collected from patients with squamous cell carcinoma of the larynx at baseline, 6, 12 and 24 months. Multiple logistic regression analyses will be used to quantify locoregional control and identify factors associated with control overall and by treatment modality and identify factors associated with quality of life overall and by treatment modality.

**Ethics and dissemination** Most interventions take place as part of routine care, with LARCH providing a mechanism for recording this data centrally. When successfully recruiting in the North of England, we plan to roll out LARCH nationwide; in the future, LARCH can be used as a trial platform in the disease. The results will be submitted for publication in high-impact international peer-reviewed journals and presented to scientific meetings. Access to the anonymised LARCH dataset by other researchers will be publicised and promoted.

**Trial registration number** ISRCTN27819867.

### STRENGTHS AND LIMITATIONS OF THIS STUDY

⇒ This is a pragmatic observational study, collecting real-world data, for which most laryngeal cancer patients presenting to participating centres will be eligible.
⇒ The specific data collected will have value in predicting response to treatment measured both in terms of clinical and patient-reported outcomes; this has not been collected before on this scale for laryngeal cancer.
⇒ Research processes allow the routine collection of radiological and tissue data, linked to patient-reported and clinical outcomes.
⇒ While the effectiveness of treatments is being compared, and rigorous epidemiological methods will be used to control confounders (such as propensity score matching), some care will be needed in interpretation of findings due to the observational design.

## INTRODUCTION

Laryngeal cancer affects 2400 new patients each year in the UK. Incidence increases with age and the cancer disproportionately affects those who are socioeconomically disadvantaged.[1] Around half of patients present with advanced disease and, consequently, have poor survival. Those affected are often left with a significantly diminished quality of life: treatment can render the individual permanently nil by mouth due to swallowing problems or in need of a permanent tracheostomy or tracheal stoma. The impact on quality of voice or communication can prevent effective communication. Swallowing difficulty and dry mouth can necessitate permanent gastrostomy tube feeding with effects on socialising and relationships. Laryngeal cancer comprises early (T1 and T2) and advanced (T3 and T4) cancer. Progress in the treatment of both has stalled over the past 30 years. There has been a sea-change in treatment algorithms, with increasing use of transoral laser in early disease and the advent of 'laryngeal preservation' in advanced disease. But, over the same time period, survival has remained at best static,[2 3] and at worst it has declined.[4] From a wider societal perspective, laryngeal cancer diagnosis and treatment costs the National Health Service (NHS) £96 million per year with estimated productivity losses of £230000/patient.[5]

The last reported randomised controlled trial (RCT) which successfully recruited patients and compared surgical and non-surgical treatments reported in 1991[6]; subsequent RCTs have either compared non-surgical treatments alone or failed to recruit successfully.[7 8] We have investigated the possibilities of clinical trials in this field, and have recognised the work done to attempt to select patients on the basis of response to up front (induction) chemotherapy.[9] However, induction chemotherapy has repeatedly shown decreased disease control rates.[7] Also, there is no consensus on how tumour response should be measured or defined in RCTs. Moreover, researchers' and clinicians' outcome priorities of locoregional control and organ preservation are not necessarily shared by the patient population; indeed, our group has shown that patients will prioritise swallow and voice function over treatment modality or survival.[10] Currently, clinicians have no method of prognosticating how a specific tumour or patient will respond to treatment. For example, one patient with advanced disease treated with radiotherapy will achieve good locoregional control, preserve their larynx, achieve a functional swallow and produce their voice normally. Another patient with the same stage of disease, treated with the same modality, may have persistent disease after 6 weeks of treatment, or will be rendered nil by mouth, gastrostomy fed and dependent on a breathing tube. This has led to clinicians relying on experience and anecdote to guide and support patients though complex decisions. This, in turn, has led to huge variation in treatments delivered across the UK; for example, the rate of primary radiotherapy for advanced laryngeal cancer varies across centres from 0% to 83%.[11]

To move forward, we require an understanding of how an individual patient with laryngeal cancer responds to treatment, with respect to both tumour control and functional outcome. Further, clinicians and researchers need to be able to pre-determine this response at presentation. Knowledge of the factors which dictate response to treatment would allow researchers to identify the variables to control for when designing future RCTs of novel treatments. However, risk prediction or precision medicine in laryngeal cancer remains elusive. Clinical cohorts, enriched with rich clinical data and biological samples, are an invaluable resource for advancing precision medicine. In head and neck cancer, audits and previous cohort studies have examined, to some extent, processes of care and survival.[12 13] This has allowed some initial hypothesis generation; however, ambitions to include patients with all cancers have limited the usefulness of the data and clinical engagement. The largest such cohort, the 'Head and Neck 5000 study' (HN5000), collected longitudinal data on 5000 patients (1065 with laryngeal cancer) with 3-year survival and patient-reported outcomes, but clinical data and samples are limited. However, the endeavour does establish proof of concept and a blueprint for a specific LARyngeal Cancer coHort (LARCH).

In other cancers sites (breast, prostate, lymphoma), biochemical analysis of biopsy samples allows the tailoring of specific treatments to an individual tumour; unfortunately in laryngeal cancer researchers have not made such progress. This is partly due to the heterogeneous tumour biology of carcinogen-induced malignancies and partly because this cancer attracts little research interest and spending (0.8% of cancer research spend in 2018/2019).[14] There are potential prognostic markers under investigation: either patient based (tobacco use, muscle mass) or tumour driven (the immune checkpoint inhibitor complex IDO, 'immunoscore', PDL-1). Although the latter are under investigation for response to novel chemotherapeutic agents, their presence does not currently influence the choice of primary treatment.[15] The emerging field of 'radiomics' uses the raw data from routine CT scans and maps this to tumour and patient outcomes as a 'radiological biomarker'.[16] This means that although every patient with laryngeal cancer undergoes tissue biopsy and cross-sectional imaging, much of the data goes to waste.

The LARCH—a cohort study of all laryngeal cancers, enriched with samples and detailed clinical data—is a major step towards defining precision medicine in laryngeal cancer and, in the longer term, to improving patient outcomes. It will allow significant progress in areas including: comparative treatment effectiveness research; investigation of biochemical and radiological biomarkers (mapped to treatment outcome); mapping clinical and patient characteristics to outcome; patient involvement in decision making; and the identification of variables for further investigation in trials of novel prognosticators or therapeutics.

### Primary objectives
To establish a disease database of laryngeal cancer patients in order to:
1. Assess the difference in quality of life, disease-specific and overall survival between treatment modalities in early and advanced laryngeal cancer.
2. Assess the impact of patient-derived clinical features and tumour factors on treatment outcome (oncological, laryngeal function, quality of life, swallow, voice) in early and advanced laryngeal cancer and use this to develop a risk prediction tool.

### Secondary objectives
1. To establish consent processes to allow researchers to recontact patients for data on long-term outcome and survivorship.
2. Using the data, establish an initial risk communication tool in the disease.
3. To develop the pathway for routine tissue and radiological scan collection for future studies, mapped to outcome.

### METHODS AND ANALYSIS
This study is the first enhanced laryngeal cancer disease cohort. We aim to deliver a prospective cohort study of a minimum of 150 patients recruited over a 3-year period.

This initial phase of the study will recruit patients from eight head and neck cancer centres in the North of England: Newcastle, Liverpool, Sunderland, Middlesbrough, Hull, Sheffield, Leeds and Manchester. Once established, in future research, we aim to roll out the cohort nationally.

This study will be pragmatic, involving patients with laryngeal cancer attending hospital as part of their routine care. Patients will be identified by a member of the research team and recruitment will be supported by the research team, all of whom will be trained in Good Clinical Practice and consent.

## Eligibility criteria
### Inclusion criteria
► Suspected but unconfirmed laryngeal cancer (group 1).
► Confirmed new diagnosis of laryngeal cancer (group 2).
► Age over 18.
► Capacity to consent.
► Ability to understand written and spoken English.

### Exclusion criteria
► Recurrence or second head and neck primary cancer.

Patients from group 1 who are ineligible for inclusion into group 2 will be withdrawn from the study and subjects replaced. Patients who withdraw from the study during the enrolment period will be replaced. Withdrawn subjects will be replaced until at least 150 eligible patients have completed the study endpoints.

### Group 1: suspected laryngeal cancer
Patients with a suspected diagnosis of laryngeal cancer will be identified by the clinical team at the time that laryngeal cancer is suspected. The patient will be given the participant information sheet (PIS) by the clinical team at the time of the clinic appointment or sent a copy of the PIS by a research nurse after the clinic appointment has taken place. On the day of the biopsy, they will be approached by a member of the research team and given an opportunity to ask any questions about the study. If the biopsy does not confirm cancer, the patient will be withdrawn from the study and any data collected so far will be destroyed. Patients with a biopsy confirming laryngeal cancer will be eligible to continue in the study and move to group 2.

### Group 2: confirmed new laryngeal cancer diagnosis
Patients who have laryngeal cancer confirmed by biopsy who have not been enrolled into the study previously will be identified by clinical staff at the clinic following their diagnosis of laryngeal cancer by biopsy. At the time of diagnosis, they will be given the PIS, consent form and the preliminary questionnaires.

## Consent
Potential participants will be assessed by a member of the study team, eligibility will be confirmed and baseline assessments performed. Consent to enter the study will be sought from each participant only after a full explanation has been given and the PIS has been given. We have worked with our patient and public involvement (PPI) representatives to ensure that these materials are concise, clear and easy to understand.

The patient is free to refuse participation without giving reasons and without this affecting the care they receive. The participant is free to withdraw at any time from the study without giving reasons and without prejudicing their further treatment. Participants are provided with a contact point where they may obtain further information about the study in the PIS. Data and samples collected up to the point of withdrawal will be kept.

Consent to additional elements of the research will be driven by patients' wishes giving them control over how their data are used. Patients will be offered the opportunity to consent to the following without affecting their inclusion within the core aspects of the study:
► Consent for future contact with information about studies that become available.
► Consent for future contact for collection of additional information regarding this study.
► Use of anonymised information collected from this study to be used in future studies which have received separate ethical approval.
► All patients who are identified before their biopsy, have a cancer recurrence or who have a resection will be eligible for consent for storage of their tissue. This consent details that their tissue will be stored indefinitely and can be used for future, as yet undecided studies. Patients have the option to give consent for further biopsies (often in the case of recurrence) to be collected and transferred to a biobank for use in future research projects.

Patients may consent to as many or few additional elements as they wish.

## Data collection
The following data will be collected on all patients; additional study questionnaires will be collected at any time before the patient begins treatment for their laryngeal cancer (table 1).

Patients will be followed up until 24 months (±3 months) after the completion of their last treatment, or until date of death if this occurs during the study. Frequency of follow-up will follow standard care (approximately 6, 12 and 24 months), with clinical and functional information collected at each follow-up visit (table 2).

If a patient has a scan as part of their routine care, the scan will be stored and the radiological data will be reported, analysed and collected as described

If a patient has a cancer recurrence, as well as the radiological data outlined above, the tissue will be collected and stored in a biobank as discussed in the consent section above. In the event of a recurrence, the data collected will be the same as a primary disease presentation.

**Table 1** Data collected at baseline assessment

| Clinical | Radiological | Functional | Histology | Treatment received |
|---|---|---|---|---|
| Age | Laryngeal subsite | EORTC QLQ-C30 | From biopsy | Primary non-surgical treatment <br> ► Chemotherapy agent and dose <br> ► Radiotherapy dose and fractionation |
| Date of birth | Cartilaginous framework involvement | EORTC HN35 | Histology | Primary surgical management-procedure performed |
| Gender at birth | Maximum dimension | Vocal cord function | Grade | Postoperative chemotherapy |
| NHS no | Extralaryngeal spread | DIGEST VFSS/FEES scale | Differentiation | Postoperative radiotherapy |
| Treatment hospital | Bilateral involvement | PSS-HN | Digital photography if available | Date of completion of treatment |
| GP postcode | Tumour max SUV | MDADI | From resection (if performed) | |
| Date of biopsy | Signs of aspiration | VHI-10 | Digital photography if available | |
| Date of initiation of treatment | Cervical lymph node involvement | 100 mL water swallow test | Primary site | |
| Aim of treatment | Evidence of extra-capsular spread | Maximum phonation time | Dimension | |
| TNM stage | | GRBAS scale | Primary site | |
| Smoking history | | Voice recording | Uni/multifocal | |
| Alcohol history | | | Histology | |
| ACE score | | | Differentiation | |
| Clinical frailty scale | | | Invasiveness | |
| WHO performance | | | Invasive front | |
| Weight | | | Perineural invasion | |
| Presence of tube feeding | | | Lymphovascular invasion | |
| | | | Closest margin | |
| | | | Pathological TNM staging | |
| | | | Neck disease | |
| | | | Total no of nodes dissected | |
| | | | Total no of involved nodes | |

DIGEST, Dynamic Imaging Grade of Swallowing Toxicity; EORTC HN35, European Organization for the Research and Treatment of Cancer Head and Neck Cancer Quality of Life Questionnaire; EORTC QLQ-C30, European Organization for the Research and Treatment of Cancer Quality of Life Questionnaire; FEES, Fiberoptic Endoscopic Evaluation of Swallowing; GP, General Practitioner; GRBAS, Grade, Roughness, Breathiness, Aesthenia, Strain Scale; MDADI, MD Anderson Dysphagia Inventory; NHS, National Health Service; PSS-HN, Performance Status Scale for Head & Neck Cancer Patients; SUV, Standardised Uptake Values; TNM, Tumour, Node, Metastasis; VFSS, Videofluoroscopic Swallow Study; VHI, Voice Handicap Index.

## Storage and analysis of biological samples

Tumour tissue biopsies at diagnosis, during treatment or on relapse, will be obtained or at surgical resection at presentation, during treatment or on relapse in line with participant's consent.

The procedure for sample retrieval will depend on the nature of the sample requested and the laboratory services on site. If consent is withdrawn for issued samples by the donor, recipients will be informed of the relevant sample numbers and asked to return any unused samples

| Table 2 | Data collected at follow-up assessment |
| --- | --- |
| **Clinical** | **Functional** |
| Disease status<br>Alive without disease<br>Alive with disease<br>Death (disease related)<br>Death (non-disease related) | EORTC QLQ-C30 |
| Smoking history | EORTC HN35 |
| Alcohol history | Vocal cord function |
| ACE score | DIGEST VFSS/FEES scale |
| Weight | PSS-HN |
| Presence of tube feeding | MDADI |
| | VHI-10 |
| | 100 mL Water swallow test |
| | Minimum phonation time |
| | GRBAS scale |
| | Voice recording |

DIGEST, Dynamic Imaging Grade of Swallowing Toxicity; EORTC HN35, European Organization for the Research and Treatment of Cancer Head and Neck Cancer Quality of Life Questionnaire; EORTC QLQ-C30, European Organization for the Research and Treatment of Cancer Quality of Life Questionnaire; FEES, Fiberoptic Endoscopic Evaluation of Swallowing; GRBAS, Grade, Roughness, Breathiness, Aesthenia, Strain; MDADI, MD Anderson Dysphagia Inventory; PSS-HN, Performance Status Scale for Head & Neck Cancer Patients; VFSS, Videofluoroscopic Swallow Study; VHI-10, Voice Handicap Index.

for destruction. Results obtained from samples that have already been used for research need not be destroyed.

### Routine imaging and radiomic analysis

All scans taken as part of routine care will be stored on hospital servers (as is usual practice). If not already in Newcastle, many of these scans will be transferred to Newcastle upon Tyne Hospital NHS Foundation Trust. These scans will be subjected to radiomic analysis: scans will be extracted in DICOM format anonymised ensuring removal of all patient identifiable information.

### Statistical analysis

Patient data and information will be collected, stored and used based on patient consent, and consistent with General Data Protection Regulation (GDPR) requirements. Descriptive statistics will be used to describe the recruited cohort and data completion (at baseline and at follow-up). Baseline characteristics will be compared with routine data (from Hospital Episode Statistics) to determine whether the recruited sample is representative of patients with laryngeal cancer across NHS in England.

Depending on the maturity of the cohort at the conclusion of the study, data will be used to start to develop risk prediction models and compare outcomes by treatment modality. Multiple logistic regression analyses will be used to (1) identify patient-related, clinical and health service-related (eg, institution) factors associated with receipt

of surgical versus non-surgical treatment; (2) quantify locoregional control and identify factors associated with control overall and by treatment modality and (3) identify factors associated with quality of life overall and by treatment modality. Epidemiological approaches to support treatment comparisons within observational datasets (eg, propensity scores, instrumental variable analysis) will be used. These analyses will provide an early demonstration of the value and potential of this new cohort.

The planned analyses will draw on a range of epidemiological analytical approaches. They will explore utility of high dimensional propensity scores (which incorporate additional variables in the propensity score, such as clinician and/or hospital characteristics, with these serving as proxies for unmeasured confounders) and instrumental variable analysis (which relies on the existence of an 'instrument', a variable that is related to the treatment but not to the study outcome other than through treatment effects).

## ETHICS AND DISSEMINATION

The study will be conducted in accordance with the recommendations for physicians involved in research on human subjects adopted by the 18th World Medical Assembly, Declaration of Helsinki 1964 and later revisions. The majority of the interventions delivered take place as part of routine care, with LARCH providing a mechanism for recording this data. Biological material will be stored under each establishment Human Tissue Authority licence. Ethical approval for the project has been obtained from London—Surrey Borders Research Ethics Committee, reference 22/PR/0406. The project has been included in the NIHR portfolio, CPMS ID: 52 643. All potential future studies will have separate ethical approval.

### Confidentiality and data management

The Caldicott Principles and GDPR will be fully adhered to when dealing with patient identifiable data. Within each recruiting site the principal investigator (PI) will preserve the confidentiality of participants taking part in the study. Within sites, no staff beyond the usual care team and local research team will have access to identifiable data. The data will be held at the site in accordance with local Trust policies and will be destroyed following the study close in accordance with local research and development protocols.

The data controller (sponsor) will have access to the full dataset but other users will be restricted to data from their own centre. Access will be via secure portal. Local sites will have access to their site data.

A purpose-built database and web portal will be developed to support the capture of patient recruitment and study participation data. Only authorised individuals will have login credentials to input this data. The staff (including research nurses within recruiting centres) will be able to remote access the portal to transfer the data.

Staff will be a delegated out by PI based on skills, training and experience; these will be used to restrict access to confidential or otherwise sensitive data. Our web portal will be deployed using REDCap is a browser-based electronic case report form, suitable for Newcastle-led non-commercial research studies. This system site is on the Health and Social Care Network (HSCN:ie, the data are securely behind the NHS firewall).

Data generated by the research project will be retained after publications resulting from the research are finalised. Data will be stored for 20 years. After 20 years data will be considered for longer-term retention based on the published results and further advances in the field of research internationally. Datasets stored for >20 years will be anonymised.

At the end of the study, strategies to promote and publicise the cohort and processes for gaining access to data, samples or the recruitment network will be developed and widely publicised and promoted. These will include: information on the cohort website; designated contact point for further information or support; flyers, stalls or sessions at scientific and stakeholder conferences; webinars or podcasts, and providing a Digital Object Identifier created through data.ncl for inclusion on the project website and published papers.

The data sharing policy, and the procedures (including application forms) for requesting access to the cohort, will be made available on the cohort website. Currently it is anticipated that applications will be prioritised based on: quality; alignment with our vision; potential to translate into clinical practice or deliver significant patient or public benefit; collaborative approach; and added value. The cohort research committee will have responsibility for reviewing and approving/declining requests for access.

Data held in the cohort database will be offered for sharing once: (1) the principles on which applications will be reviewed, prioritised and approved are agreed within the collaboration; (2) the processes for applying to access the cohort are agreed and implemented; and (3) reasonable numbers of patients have been recruited to the cohort. Investigators of adopted research studies will develop their own timelines for data sharing. A requirement of cohort approval will be that data are made available for sharing at as early a point as possible. For these studies, we currently anticipate that the applicant, research partner and other partners and collaborators will have access to the data for 1–2 years after the end of data collection for the study. Patients recruited to the cohort will be asked for consent to data sharing. For the cohort-adopted studies, it will be a requirement of adoption that consent is obtained from recruits for data sharing. The major delay to data sharing which will occur relates to the need to establish procedures for study review and adoption

The study management group (SMG) will make the decision on data access. Committee membership will include the chief investigator (DWH) and coinvestigators (JO'H and LS) and PPI representation. This group will act as gatekeepers for the data. Decisions on whether or not an application is approved will be recorded together with, in the event the application is declined, reasons and whether the applicants may reapply if they fulfil certain conditions.

The Newcastle upon Tyne Hospitals NHS Foundation Trust, as data controller, will require data sharing agreements with all organisations responsible for submitting and accessing data.

Patients will have their date of birth and NHS number collected and this will be stored in a secure purpose-built database. Where data linkage to national databases such as NHS Digital, National Cancer registries or Hospital Episode statistics is required, this identifiable information will be passed onto these national organisation to allow data linkage.

Patients who have withdrawn consent will have all data collected up until the point of withdrawal included in the study. These data will be uploaded onto an electronic case report form (CRF) and included in the analysis of the study. Data will be submitted either directly onto the electronic CRF or onto paper CRF before input into the electronic CRF. In relation to consent for contact, researchers undertaking studies requiring access to identifiable data (eg, in order to contact patients about participation in a new research study) will be granted appropriate access on the basis of agreement from the study management team and subject to ethical approval and consistent with patient consent.

## Use of tumour samples or data by other researchers

Researchers may request to use material from the LARCH study which have been stored in Biobanks for future research. Researchers will be required to have their own NHS research ethics committee approval for their research projects, and the release of the material must be approved by the LARCH access committee to ensure an appropriate use of samples. This will be conducted in accordance with a formalised Access Policy and procedure. Access to the tissue collection is available to research groups based in the UK and elsewhere.

## Patient and public involvement

We explored multiple potential study designs during the preparation of this protocol with three PPI groups; these groups were not in favour of a randomised trial design. In defining precision medicine in laryngeal cancer, we do not aim to show whether a particular treatment is 'better' for patients. Rather, we aim to specify the tumour and patient factors which lead to a poor response to laryngeal cancer treatment (in general, and for specific treatments) in individual patients. PPI groups have checked and approved all patient facing study documents. We have a patient representative on the SMG and a patient (GS) is a coauthor of this protocol and was involved in manuscript preparation

## Sponsor

LARCH will be led by Newcastle University with The Newcastle upon Tyne Hospitals NHS Foundation Trust acting as sponsor. In participating sites, a PI will be identified.

Portfolio adoption will ensure that NIHR CRN research nurses are able to undertake patient recruitment.

## Study management

LARCH will be coordinated by an SMG. The SMG will be chaired by the chief investigator and comprise coinvestigators, PPI representation, study manager, research fellow, lead nurse, data managers and digital research engineers. The SMG will advise on matters including appropriateness and sensitivity of patient-facing materials; methods of patient approach, consent and data collection; strategies to maximise retention; interpretation of results; and dissemination to lay audiences. Active involvement of the will help maximise patient acceptability and, hence, recruitment and retention. The study management team will govern all access to data.

A representative group of staff who will be part of the research team at each site will attend a site initiation meeting to ensure compliance with the protocol and allow training in study procedures and data collection methods. The PI at each study site will apply for local research and development (R&D) approval. The PI will sign a copy of the ethically approved protocol to confirm agreement to carry out all study related tasks in accordance with the protocol. Deviations from protocol will be reported to the SMG.

## Dissemination

Data from the study will be disseminated to participating centres within the study via the Trust's websites. Additionally, lay summaries will be prepared, posted on the study website and disseminated through websites of individual participating sites, charities and locations advised by PPI. Digital tools for dissemination of research findings will be developed and information disseminated to individuals requesting this.

For academic and clinical dissemination, the results will be submitted for publication in high-impact international peer-reviewed journals and presented to scientific meetings.

**Contributors** All authors have made substantial contributions to the conception or design and drafting of the work. They have all approved of the version to be published and agree to be accountable for all aspects of the work. Authors' specific contributions are detailed as follows: DWH: corresponding author, research question, research design, manuscript preparation, lead on revisions. JO'H: research question, research design, manuscript revisions. AR: research question, research design, manuscript revisions. GS: research design and conduct. MA: research governance, quality assurance, manuscript revision. KK: research governance and safety, manuscript revisions. RP: research governance and safety, manuscript revisions. DT: analysis strategy, manuscript revisions. JP: research design, outcome measures, manuscript revision. TMJ: research question, research design, manuscript revisions. LS: research question, research design, manuscript revisions.

**Funding** Funding to this study is from The MRC/NIHR Clinical Academic Research Partnership (CARP; grant number MR/V037528/1), and is on NIHR portfolio (CPMS: 52643) gaining additional support in identifying patients and gaining consent.

**Competing interests** None declared.

**Patient and public involvement** Patients and/or the public were involved in the design, conduct, reporting and dissemination plans of this research.

**Patient consent for publication** Not applicable.

**Provenance and peer review** Not commissioned; externally peer reviewed.

**ORCID iDs**
David Winston Hamilton http://orcid.org/0000-0002-9653-6453
Rosalyn Parker http://orcid.org/0000-0002-0873-6878
Linda Sharp http://orcid.org/0000-0001-9515-1722

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
