## [Reviewer comments · BMJ Open]

ARTICLE DETAILS

TITLE (PROVISIONAL)	Precision medicine in laryngeal cancer: protocol of the laryngeal cancer cohort (LARCH)
AUTHORS	Hamilton, David; O'Hara, James; Rajgor, Amarkumar; Selby, Gerald; Anderson, Mhairi; Keltie, Kim; Parker, Rosalyn; Teare, Dawn; Patterson, Joanne; Jones, Terry; Sharp, Linda

VERSION 1 – REVIEW

REVIEWER	Anna Rzepakowska Medical University of Warsaw, Department of Otorhinolaryngology, Head and Neck Surgery
REVIEW RETURNED	28-Oct-2022

GENERAL COMMENTS	The article presents a protocol for a cohort study of patients with laryngeal cancer, initially in the North of England, eventually nationwide. The desirability of such analyzes is thoroughly discussed by the authors in the introduction and I fully share the view of the lack of progress in the treatment methods for this group of patients, the lack of individual therapeutic approaches, heterogeneous methods of evaluating functional results post the treatment and the still unsatisfactory results of molecular tests. All these aspects translate into a huge variety of treatments used locally and worldwide. The joint collection of clinical and radiological data and tissue samples is fully justified, and the possibility of using the material for future researches creates opportunities for a thorough analysis of factors influencing survival and indication of optimal, individualized treatment methods. The protocol, after appropriate adaptation to local regulations, should be disseminated as soon as possible in Europe and other countries in order to implement the objectives as quickly as possible, taking into account geographical and ethnic differences.
---

REVIEWER	Patrick Bradley University of Nottingham, ORL-HNS
REVIEW RETURNED	01-Nov-2022

GENERAL COMMENTS	This manuscript describes how members with others of the Head and Neck Department based at Newcastle upon Tyne NHS Foundation Trust propose to conduct a prospective cohort study on 150 patients suspected and diagnosed with laryngeal cancer in eight centres over a period of 3 years assessing the differences in outcomes (QoL, function, survival etc) between different modalities in early and late cancer. 1) Abstract -- there is a need to define the histopathological subtype
---

of cancer as being squamous cell.

2) As eight centres in the North of England - these Centres should be listed if already have a agreed to participate! A Head and Neck Centre has been loosely defined in the UK as diagnosing and treating more than 100 new Head and Neck Cancers per year!

3) Mention is made that after recruiting the 150 "pilot" study that the project will be rolled-out nationwide --- also stated that the data would be prospectively collected but retrospectively analysed!

4) What if a Centre is not recruiting to expectations ---- will the Centre be deleted and or replaced?

5) While the application of radiomics to the data has been introduced and may be of help in analysis of advanced stage disease and the best selection of treatment surgery vs (chemo)-radiotherapy the reference 11 is of a UK report using only 15 patients !

6) In the strength and limitations of the study (page 4) "While the effectiveness of treatments are being compared, and rigorous methods will be used to control confounders" this should be explained further!

7) Introduction: First paragraph, 4th last line "But, over the same period ..." is there really any UK evidence to show this, even England and Wales?

8) Introduction: First paragraph 3rd last line "From a wider social" this is an Irish group that analysed Irish retrospective data and reported in Euros --- can this really be a substituted for the UK population and translated into sterling £?

9) Introduction: Page 5, Second paragraph "Tumour response (TR)" within this project how is TR to be measured or assessed -- and in the main complete response is the only meaningful response that has significant meaning!

10) Introduction: Page 5, Second paragraph, 3rd last line "This, in turn" this is the last and most recent report on treatment and survival outcome in the UK on 2013 Data -- that's 9 years ago? The National Database has not been replaced -- so the divergence of treatment options for early stage glottic cancer may have changed -- BUT this data is National Data

11) Introduction: Page 6, "Head and Neck 5000 Study" refers to data collected based in Bristol which continued to be analysed --- the larynx data collected (1065 larynx cancer) has been approved for analysis and reporting in September 2021 by the Lead Author David Hamilton) . Some of the data has already been reported by O'Hara J et al on T1a glottic cancer (123 patients) which showed that the treatment was almost equal between surgery (55) vs radiotherapy (68) with equivalent survival outcomes (The contributors to this data pool was selected UK head and neck Cancer centres!

12) Assessment of QoL should in reality be a comparison between before diagnosis and after treatment of a patients cancer! How can this be entered into the data collection!

13) Methods and analysis (Page 7) there should be more information on the training of the "research nurses" who approach and discuss with patients about their likely diagnosis, trial etc -- to avoid persuasion, rejection etc

14) As all stages of larynx cancer are being recruited -- should this project consider that the early -- surgery vs non-surgery be separated from late - surgery vs non-surgery with equal patient numbers in both groups?

15) Seems a shame that patients data whom refuse/ withdraw from such a study (mainly Stage 2) should be reported as to the reasons why and reported -- may vary between regions or districts in the UK, or other reasons! The Group has already had experience with the TUBE Trial in the past!

	16) Assessments Page 9/10 the Functional Assessment Baseline and Follow-Up is comprehensive and desirable how is it envisaged that such assessments will be conducted mainly by speech and language therapists (SLT)? Will a research clinic/session be funded for such? Has each of the 8 Centres been adequately staffed with SLT? A ambitious project, worthy of funding, requires early recruitment in the first year, with prospective data reporting of data to convince participants that completion can be achieved!
--	--

VERSION 1 – AUTHOR RESPONSE

Reviewer: 1

Dr. Anna Rzepakowska, Medical University of Warsaw Comments to the Author:

Thank you for the review. There were no specific points to address in the protocol, and thus, no changes were made to the manuscript, based on this review

Reviewer: 2

Dr. Patrick Bradley, University of Nottingham Comments to the Author:

- 1) **Abstract -- there is a need to define the histopathological subtype of cancer as being squamous cell.** The statement “collected from patients with squamous cell carcinoma of the larynx” has been added to the abstract
- 2) **As eight centres in the North of England - these Centres should be listed if already have a agreed to participate! A Head and Neck Centre has been loosely defined in the UK as diagnosing and treating more than 100 new Head and Neck Cancers per year!** At the start of the methods section (page 7), the statement “Newcastle, Liverpool, Sunderland, Middlesbrough, Hull, Sheffield, Leeds and Manchester.” has been added
- 3) **What if a Centre is not recruiting to expectations ---- will the Centre be deleted and or replaced?** As the study is on the NIHR Portfolio, centres will be required to recruit at a planned rate in order to meet time and target recruitment figures. whether a centre stays within the study when they have low recruitment is a decision for the centre based on portfolio recruitment figures rather than the Trial Management Group
- 4) **In the strength and limitations of the study (page 4) "While the effectiveness of treatments are being compared, and rigorous methods will be used to control confounders" this should be explained further!** This statement has now been changed to “While the effectiveness of treatments are being compared, and rigorous epidemiological methods will be used to control confounders (such as propensity score matching), some care will be needed in interpretation of findings due to the observational design
- 5) **Introduction: First paragraph, 4th last line "But, over the same period ..." is there really any UK evidence to show this, even England and Wales?** References have been added to this statement: first, Rachet, 2008, which shows static survival between 1986 and 2001, and secondly, the CRUK data from the present day, demonstrating survival equivalent to 1986. The reference to show a decline is Hoffman, admittedly taken from the US, rather than the UK, but we have kept the statement in as I think it highlights a concerning point
- 6) **Introduction: First paragraph 3rd last line "From a wider social" this is an Irish group that analysed Irish retrospective data and reported in Euros --- can this really be a substituted for the UK population and translated into sterling £?** Although this is Irish data, we feel that the quality of the analysis demonstrates a point which can be extrapolated beyond Ireland alone. The senior author on this work (LS) is the senior author on this protocol

- 7) **Introduction: Page 5, Second paragraph "Tumour response (TR)" within this project how is TR to be measured or assessed -- and in the main complete response is the only meaningful response that has significant meaning!** We agree with this statement; within this specific paragraph we are discussing our previous attempts to progress laryngeal cancer management through bioselection; a partial, complete or absent response to induction chemotherapy in this situation may allow us to guide treatment options, but there is no consensus as to how this should be measured. In the present study, tumour response is determined clinically; as treatments are not being altered patients will be treated and followed up according to local protocol and there will be no deviation from standard treatment pathways
- 8) **Introduction: Page 5, Second paragraph, 3rd last line "This, in turn" this is the last and most recent report on treatment and survival outcome in the UK on 2013 Data -- that's 9 years ago? The National Database has not been replaced -- so the divergence of treatment options for early stage glottic cancer may have changed -- BUT this data is National Data** The lack of national survival statistics over the last 9 years is indeed worrying and the authors are gathering these data for publication next year. Prof Jo Patterson (co-author of this work) is currently leading a nationwide audit of outcomes after treatment for early laryngeal cancer (albeit in limited centres which geographically span the UK).
- 9) **Introduction: Page 6, "Head and Neck 5000 Study" refers to data collected based in Bristol which continued to be analysed --- the larynx data collected (1065 larynx cancer) has been approved for analysis and reporting in September 2021 by the Lead Author David Hamilton) . Some of the data has already been reported by O'Hara J et al on T1a glottic cancer (123 patients) which showed that the treatment was almost equal between surgery (55) vs radiotherapy (68) with equivalent survival outcomes (The contributors to this data pool was selected UK head and neck Cancer centres!** The analysis of the advanced laryngeal cancer data is ongoing and will be submitted for publication next year. James O'Hara, who demonstrated equal quality of life outcomes between surgery and radiotherapy in early laryngeal cancer, is named as a co-author on this protocol
- 10) **Assessment of QoL should in reality be a comparison between before diagnosis and after treatment of a patients cancer! How can this be entered into the data collection!** LARCH will routinely collect all quality of life data at baseline (ie before the start of treatment) and again at 6, 12 and 24 months
- 11) **Methods and analysis (Page 7) there should be more information on the training of the "research nurses" who approach and discuss with patients about their likely diagnosis, trial etc -- to avoid persuasion, rejection etc** As LARCH is on the NIHR portfolio, there is an expectation that all involved research nurses will be part of NIHR CRN; as such they will have experience of, and be trained in, consent practices. To add clarity, we have added the statement "all of whom will be trained in Good Clinical Practice and consent."
- 12) **As all stages of larynx cancer are being recruited -- should this project consider that the early -- surgery vs non-surgery be separated from late - surgery vs non-surgery with equal patient numbers in both groups?** The analysis of early and late stage cancer will be separated. However, as this is a pragmatic cohort study, there have been no limits or guides placed on recruitment

- 13) Seems a shame that patients data whom refuse/ withdraw from such a study (mainly Stage 2) should be reported as to the reasons why and reported -- may vary between regions or districts in the UK, or other reasons! The Group has already had experience with the TUBE Trial in the past!** Collecting data from those who refuse to consent to a cohort study which essentially collects data is problematic and was considered by the trial team to therefore be impossible. We are not anticipating a large number of refusers. The lead author (DH) was involved in the pilot and PPI work for the TUBE trial which randomised patients between treatment options. As no randomisation or treatment changes are being made in LARCH, we expect that we will not encounter the same difficulty
- 14) Assessments Page 9/10 the Functional Assessment Baseline and Follow-Up is comprehensive and desirable how is it envisaged that such assessments will be conducted mainly by speech and language therapists (SLT)? Will a research clinic/session be funded for such? Has each of the 8 Centres been adequately staffed with SLT?** This is a vital point: engagement of SALT is central to the success of LARCH. There is currently no formal funded time for SALT, but all of the measures are collected as part of routine practice. There is currently a national SALT audit, collecting very similar data in early larynx cancer and the lead author (DH) attends the meetings of this group. LARCH has enrolled on the Associate PI (API) scheme, and the first API in place is a speech and language therapist; we will be encouraging all centres to do the same. One of the co-authors (JP) is a professor of speech and language therapy. Lastly we have applied to the NIHR “Targeted Health Needs” to fund some SALT time to ensure adequate data collection for these patients. However, the majority of the outcome measures (quality of life questionnaires, water swallow test) can be delivered by research nurses, and remotely if required

VERSION 2 – REVIEW

REVIEWER	Patrick Bradley University of Nottingham, ORL-HNS
REVIEW RETURNED	22-Dec-2022
GENERAL COMMENTS	Thanks for responding in an appropriate and speedy manner to the many concerns (total 14) which I previously expressed and am satisfied with the answers provided.